# 🐉 Baize: An Open-Source Chat Model with Parameter-Efficient Tuning on Self-Chat Data

**Canwen Xu**[1*], **Daya Guo**[2*], **Nan Duan**[3], **Julian McAuley**[1]

[1]University of California, San Diego, [2]Sun Yat-sen University, [3]Microsoft Research Asia

[1]{cxu,jmcauley}@ucsd.edu, [2]guody5@mail2.sysu.edu.cn,
[3]nanduan@microsoft.com

## Abstract

Chat models, such as ChatGPT, have shown impressive capabilities and have been rapidly adopted across numerous domains. However, these models are only accessible through a restricted API, creating barriers for new research and progress in the field. We propose a pipeline that can automatically generate a high-quality multi-turn chat corpus by leveraging ChatGPT to engage in a conversation with itself. Subsequently, we employ parameter-efficient tuning to enhance LLaMA, an open-source large language model. The resulting model, named Baize, demonstrates good performance in multi-turn dialogues with guardrails that minimize potential risks. Additionally, we propose a new technique called Self-Distill with Feedback, to further improve the performance of the Baize models with feedback from ChatGPT. *The Baize models and data are released for research purposes only.*[1]

## 1 Introduction

The rapid advancement of natural language processing (NLP) techniques in recent years has led to the emergence of highly capable chat models, such as LaMDA (Thoppilan et al., 2022), ChatGPT (OpenAI, 2023a) and GPT-4 (OpenAI, 2023b). These models demonstrate a remarkable ability to understand and generate human-like responses in a wide range of domains. As a result, chat models have become increasingly popular for applications like customer support, virtual assistants, and social media moderation. Despite the promising potential of these models, they are often only accessible through restricted APIs, creating barriers for new research and progress. Furthermore, the limited availability of chat models poses obstacles for researchers and practitioners, hindering the growth of the NLP community. The lack of publicly available,

high-quality chat corpora for multi-turn conversations exacerbates this issue, limiting the possibilities for refining and evaluating these models.

In this paper, we propose a novel pipeline (shown in Figure 1) to address these challenges by leveraging the capabilities of ChatGPT to automatically generate a high-quality multi-turn chat corpus. Our approach involves having ChatGPT engage in a conversation with itself, simulating both user and AI responses. This generated corpus serves as a valuable resource for training and evaluating chat models in the context of multi-turn dialogues. Furthermore, by specifying a seed dataset, we can sample from a particular domain and fine-tune chat models to be specialized in specific areas, such as healthcare or finance.

To fine-tune large language models in a low-resource setting, we utilize a parameter-efficient tuning approach that effectively leverages the limited computational resources available. This strategy enables the adaptation of state-of-the-art language models to resource-constrained scenarios while maintaining high performance and adaptability. Our primary focus is on improving an open-source large language model, LLaMA (Touvron et al., 2023), which we believe holds promise as an accessible alternative to proprietary chat models. By fine-tuning LLaMA with our generated chat corpus, we create a new model, named **Baize** (Bái zé, a mythical creature in Chinese folklore, who speaks human languages and knows everything). Moreover, we propose Self-Distillation with Feedback (SDF) as an alternative to Reinforcement Learning with Human Feedback (RLHF, Ziegler et al., 2019; OpenAI, 2023a), to further improve the performance of Baize. Baize is a chat model that can run on a single GPU, making it accessible for a broader range of researchers.

To summarize, our main contributions in this paper are as follows:

- We propose a novel and reproducible pipeline

---

*Equal contribution.

[1]https://github.com/project-baize/baize-chatbot

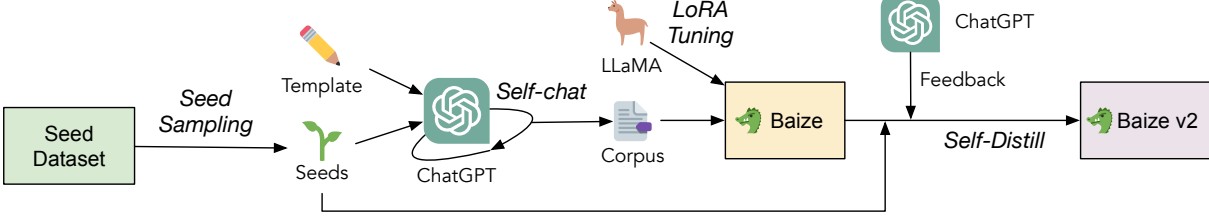

Figure 1: The pipeline for training Baize and Baize v2.

for automatically generating a high-quality multi-turn chat corpus by having ChatGPT engage in a conversation with itself. Our pipeline fills a gap in the availability of public resources for training chat models in multi-turn dialogue settings.

- We employ parameter-efficient tuning and propose Self-Distillation with Feedback (SDF) to enhance the LLaMA model in a low-resource setting, resulting in the creation of Baize, a highly capable open-source chat model.

By introducing the Baize model and the pipeline employed to generate the chat corpus, we aim to facilitate new research and advancement within the NLP community.

## 2 Related Work

**Language Models for Chat** Since the success of GPT-2 (Radford et al., 2019), there have been many language models for chatting with humans. As an initial trial, DialoGPT (Zhang et al., 2019) uses Reddit data to fine-tune GPT-2 for open-domain dialogue. Meena (Adiwardana et al., 2020) is a multi-turn open-domain chatbot with 2.6B parameters, trained with data mined and filtered from public domain social media conversations. Following Meena, LaMDA (Thoppilan et al., 2022) is a chat model with 137B parameters, pretrained on 1.56T words of public dialog data and web text. ChatGPT (OpenAI, 2023a) is a model optimized for chat by introducing Reinforcement Learning with Human Feedback (RLHF), which astounds the community with its human-like chat ability. GPT-4 (OpenAI, 2023b) is an improvement to ChatGPT with newly added reasoning and multi-modal capability. Li et al. (2022) use in-context learning with GPT-3 to augment a dialogue dataset.

Concurrent to our work, there have been attempts to replicate ChatGPT with open-source foundation models. Stanford Alpaca (Taori et al.,

2023) uses Self-Instruct (Wang et al., 2022) to collect data from GPT-3.5 in instruction learning format. Then, the collected dataset is used to fine-tune LLaMA (Touvron et al., 2023). Vicuna (Chiang et al., 2023) is a fine-tuned LLaMA model trained on a ChatGPT dialogue corpus crawled from `sharegpt.com`, a website for sharing ChatGPT dialogues. We will discuss the pros and cons of the data source of each model in Section 3.

**Parameter-Efficient Tuning** Conventional fine-tuning requires training all parameters in a large model, which can be inefficient as the numbers of parameters grows. Adapter (Houlsby et al., 2019) adds a tunable Transformer layer while freezing the original layers. BitFit (Zaken et al., 2022) only tunes bias terms in the linear layers. Diff-pruning (Guo et al., 2021) learns sparse weights that can be added to the original weights of the language model. Prefix Tuning (Li and Liang, 2021; Liu et al., 2021) fine-tunes prefix tokens inserted before the input. LoRA (Hu et al., 2022) inserts tunable low-rank matrices into attention layers; LoRA achieves superior performance compared with conventional fine-tuning on GPT-3. Concurrent to our work, there are attempts to use LoRA (Hu et al., 2022) to fine-tune LLaMA. Alpaca-LoRA[2] follows the same recipe as Alpaca while using LoRA for higher efficiency. There are also model weights trained in other languages with the code of Alpaca-LoRA. Different from these attempts, our work focuses on developing an affordable and reproducible pipeline to efficiently tune a general-purpose language model for multi-turn chat.

## 3 Data Collection via Self-Chat

In this section, we detail the methodology employed for generating a high-quality multi-turn chat corpus by leveraging ChatGPT (`gpt-3.5-turbo`) to engage in a conversation with itself. This process, named *self-chat*, serves as the foundation of

---

[2] `https://github.com/tloen/alpaca-lora`

| Seed | How do you fix a Google Play Store account that isn't working? |
|---|---|
| **Human:** | My Google Play Store account is not working properly. How can I fix it? |
| **AI:** | There are several things you can try to fix a Google Play Store account that is not working properly. First, check your internet connection and make sure it is working properly. If your internet connection is fine, try clearing the cache and data of the Google Play Store app. If that doesn't work, try uninstalling and reinstalling the app. |
| **Human:** | How do I clear the cache and data of the Google Play Store app? |
| **AI:** | To clear the cache and data of the Google Play Store app, go to Settings, then Apps & notifications, then find the Google Play Store app and tap on it. From there, tap on Storage, then Clear Cache and Clear Data. |
| **Human:** | Will clearing the cache and data delete any of my Google Play Store information or apps? |
| **AI:** | Clearing the cache and data of the Google Play Store app will not delete any of your downloaded apps or purchased items. However, it may remove your preferences and settings within the app. |
| **Human:** | What if clearing the cache and data doesn't work? |
| **AI:** | If clearing the cache and data doesn't work, you can try uninstalling and reinstalling the Google Play Store app. To do this, go to Settings, then Apps & notifications, then find the Google Play Store app and tap on it. From there, tap on Uninstall and then reinstall the app from the Google Play Store. |

Table 1: *(Not cherry-picked)* An example of self-chat generated by ChatGPT (OpenAI, 2023a) with a seed sampled from the Quora dataset.

our data collection pipeline and plays a critical role in enhancing the open-source large language model, LLaMA, to achieve better performance in multi-turn dialogues.

The self-chat process involves utilizing Chat-GPT to generate messages for both the user and AI assistant in a conversational format. We apply a template (shown in Appendix A) to define the format and requirements, allowing the ChatGPT API to continuously generate transcripts for both sides of the dialogue until a natural stopping point is reached. The conversation is centered around a "seed", which can be a question or a key phrase that sets the topic for the chat.

In our own training of Baize, we use questions from Quora[3] and Stack Overflow[4] as seeds. A dialogue example generated with self-chat is shown in Table 1. For training the first version of Baize family (**Baize v1**), we collect a total of 111.5k dialogues through self-chat, using ∼55k questions from each source. This process cost us approximately $100 for calling OpenAI's API. Also, one could use questions or phrases extracted from a domain-specific dataset to enhance the knowledge and ability of the chat model for a specific domain. Motivated by a recent report (Johnson et al., 2023) that ChatGPT can answer cancer-related questions as well as The National Cancer Institute, we use the MedQuAD (Ben Abacha and Demner-Fushman,

| Data | Dialogs | Avg. Turns | Avg. Len. |
|---|---|---|---|
| Alpaca (2023) | 51,942 | 1.0 | 44.2 |
| Quora | 54,456 | 3.9 | 35.9 |
| StackOverflow | 57,046 | 3.6 | 36.0 |
| MedQuAD | 46,867 | 3.8 | 35.8 |
| Quora v2 | 55,770 | 3.0 | 149.6 |
| StackOverflow v2 | 112,343 | 3.9 | 78.2 |

Table 2: Statistics of the number of dialogues, average number of turns, and response lengths of each turn.

2019) dataset as seeds and obtain an additional 47k dialogues in the medical domain to train a Baize model specialized for healthcare.

Note by directly generating the dialogue with the template shown in Appendix A, ChatGPT's output of each turn seems to be shorter than asking Chat-GPT one turn at a time. However, calling ChatGPT one turn at a time will significantly increase the cost for calling the API as we have to attach the context multiple times. To collect data with better quality for training **Baize v1.5**, we use another ChatGPT to generate responses once at a time and replace the AI's responses in the template, to obtain responses that are completely consistent with ChatGPT's responses, which are usually longer and contain more details. The statistics of the resulting corpora are shown in Table 2.

**Comparison with Other Data Sources** Stanford Alpaca (Taori et al., 2023) uses Self-Instruct (Wang et al., 2022) to collect data in instruction learning format. However, their instruction-input-output

---

[3] https://huggingface.co/datasets/quora
[4] https://huggingface.co/datasets/pacovaldez/stackoverflow-questions

| Model | Base Model | Type | Param. | Trainable Param. | GPU hrs | Data |
|---|---|---|---|---|---|---|
| Baize-v1-7B | LLaMA-7B | SFT | 7B | 17.9M | 9 | Quora, Stack Overflow, Alpaca |
| Baize-v1-13B | LLaMA-13B | SFT | 13B | 28.0M | 16 | Quora, Stack Overflow, Alpaca |
| Baize-v1-30B | LLaMA-30B | SFT | 30B | 54.6M | 36 | Quora, Stack Overflow, Alpaca |
| Baize-Healthcare | LLaMA-7B | SFT | 7B | 17.9M | 5 | Quora, MedQuAD |
| Baize-v1.5-7B | LLaMA-7B | SFT | 7B | 17.9M | 32 | Quora v2, Stack Overflow v2 |
| Baize-v1.5-13B | LLaMA-13B | SFT | 13B | 28.0M | 64 | Quora v2, Stack Overflow v2 |
| Baize-v2-7B | Baize-v1.5-7B | SDF | 7B | 17.9M | 38 | Quora |
| Baize-v2-13B | Baize-v1.5-13B | SDF | 13B | 28.0M | 76 | Quora |

Table 3: Data, numbers of parameters and training time for training Baize models. The GPU hours are with NVIDIA A100-80G GPUs. Baize v1 and v2 are trained with a single GPU and v1.5 are trained with 8 GPUs.

format, introduced in T0 (Sanh et al., 2022) and FLAN (Wei et al., 2022), is limited to a single turn and differs from the natural dialogue interface of ChatGPT. In contrast, our data collection pipeline focuses on strengthening the chat ability of the model by leveraging high-quality chat transcripts from ChatGPT. Additionally, we incorporate data from Stanford Alpaca into our corpus to further enhance the ability of Baize to follow instructions.

Vicuna (Chiang et al., 2023) uses dialogues crawled from sharegpt.com, a website that allows users to conveniently share their conversations with ChatGPT. An advantage of doing so is the high quality of collected data. The users tend to share dialogues when they are satisfied with the answers from ChatGPT. However, this source may have serious privacy and legal problems. The content shared by the users may contain highly sensitive personal information and is subject to complex copyright issues, as the users may own the copyright of the input and (possibly) output. Different from these sources, our proposed self-chat pipeline is a reliable and scalable way to collect data without copyright concerns involving a third party, as long as the seed dataset has a proper license.

## 4 Model Training

**Parameter-Efficient Supervised Fine-Tuning** Standard fine-tuning often requires vast amounts of computational resources, as well as high-quality and extensive datasets. However, given the limited availability of high-quality multi-turn chat corpora, it is crucial to adopt methods that are more efficient in terms of computational cost and data requirements. Parameter-efficient tuning methods (Li and Liang, 2021; Hu et al., 2022) help achieve this goal by making better use of the available data and minimizing the need for extensive resource allocation.

Specifically, we use Low-Rank Adaption method

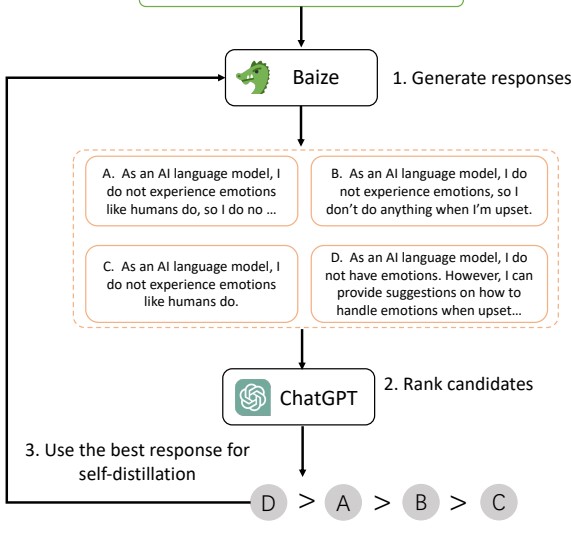

Figure 2: An overview of self-distillation with feedback from ChatGPT.

(LoRA, Hu et al., 2022) to fine-tune the LLaMA model. For a linear layer $h = W_0 x$, the forward pass is modified to be:

$$h = W_0 x + B^{sft} A^{sft} x \qquad (1)$$

where $W_0 \in \mathbb{R}^{d \times k}$, $B^{sft} \in \mathbb{R}^{d \times r}$ and $A^{sft} \in \mathbb{R}^{r \times k}$ are model parameters with the low rank $r \ll \min(d, k)$. Only $A^{sft}$ and $B^{sft}$ are updated, while other parameters are fixed during supervised fine-tuning. Different from Hu et al. (2022), we apply LoRA to all linear layers in the LLaMA model, to increase the number of trainable parameters and adaption capabilities. We list the numbers of parameters of each model in Table 3. For Baize v1.5, following Vicuna, we only compute loss for AI's responses in the dialogue transcript.

**Self-Distillation with Feedback** After supervised fine-tuning (SFT) the LLaMA model on self-chat dataset, we introduce a new way named self-

| | |
|---|---|
| **Human:** | Analyze the reason why Lehman Brothers went bankrupt. |
| **Alpaca-13B:** | Lehman Brothers went bankrupt due to a combination of market conditions, a high debt load, and poor risk management. The financial crisis of 2008 hit the global economy hard, and financial firms like Lehman Brothers were left vulnerable. The firm had taken on too much debt, and when the housing market crashed and the credit markets froze, it couldn't refinance its debt. As a result, it filed for bankruptcy in 2008. |
| **ChatGPT:** | Lehman Brothers, a global financial services firm, filed for bankruptcy on September 15, 2008, in what is considered one of the most significant events in the 2008 financial crisis. Several factors contributed to the firm's collapse, including:
- Subprime Mortgage Crisis: …
- Overreliance on Short-Term Borrowing: …
- Excessive Risk-Taking: …
- Poor Risk Management: …
- Lack of Regulatory Oversight: …
Overall, the combination of these factors led to Lehman Brothers' collapse and had far-reaching consequences for the global financial system. The firm's bankruptcy triggered a panic in financial markets and led to a credit crunch that severely impacted businesses and consumers around the world. |
| **Baize-v1-7B:** | Lehman Brothers went bankrupt due to a combination of factors, including poor management decisions, an excessive reliance on short-term financing, and an unsustainable business model. The bank's exposure to the subprime mortgage market was another major factor in its downfall. Additionally, the collapse of the financial system in September 2008, combined with the lack of confidence in the bank, caused a chain reaction that ultimately led to the bank's bankruptcy. |
| **Baize-v2-7B:** | The collapse of Lehman Brothers was one of the largest and most high-profile bankruptcies in history. Here are some of the key factors that contributed to its downfall:
- Over-leveraged balance sheet: …
- Subprime mortgage crisis: …
- Lack of liquidity: …
- Poor risk management: …
- Regulatory failures: …
- External factors: …
Overall, the failure of Lehman Brothers was a result of a combination of factors, including over-leveraged balance sheet, lack of liquidity, poor risk management, regulatory failures, external factors, and the subprime mortgage crisis. |

Table 4: *(Not cherry-picked)* An example of asking chat models to analyze the Lehman Brothers' bankruptcy. Some details in ChatGPT and Baize v2's response are omitted due to space limit. Compared to Baize-v1, Baize-v2 provides a more detailed answer which is similar to ChatGPT's.

Distillation with feedback (SDF) to self-improve the model's performance and results in Baize v2.

Figure 2 gives an overview of SDF. First, we use the resulted Baize v1.5 models to generate four responses for each instruction from the Quora dataset mentioned in Table 2. We then engage ChatGPT using the prompt provided in Appendix C to rank generate responses for self-distillation. Finally, we select the best response ranked by ChatGPT to fine-tune the model. During SDF, we apply new LoRA modules to all linear layers in Baize v1.5. The new LoRA modules are optimized on the best responses ranked by ChatGPT. For each linear layer $h = W_0 x + B^{sft} A^{sft} x$ in Equation 1, the forward pass is modified to be:

$$ h = W_0 x + B^{sft} A^{sft} x + B^{sdf} A^{sdf} x \quad (2) $$

where $B^{sdf} \in \mathbb{R}^{d \times r}$ and $A^{sdf} \in \mathbb{R}^{r \times k}$ are model parameters with the low rank $r \ll \min(d, k)$. Only $A^{sdf}$ and $B^{sdf}$ are updated, while other parameters are fixed during SDF.

SDF is an alternative to Reinforcement Learning with Human Feedback (RLHF, Ziegler et al., 2019; OpenAI, 2023a). SDF does not require training of reward models and is $3\times$ faster than RLHF, which uses PPO (Schulman et al., 2017) to optimize the model. Besides, SDF involves distillation on Baize's own generation, thus has an overall lower loss, allowing the model to capture the nuance in the feedback and perform fine-grained optimization without causing possible catastrophic forgetting. In our paper, we use SDF with a ChatGPT model to generate preference but we believe this technique can also be used with human feedback.

## 5 Model Settings

During the training phase, we set the maximum length of the input sequence to 512/1024 for Baize v1/v2 and the rank $k$ in LoRA to 8. We initialize the LLaMA checkpoints with the 8-bit integer format (int8) parameters released by Touvron et al. (2023), which remain fixed during training, thus reducing

| | |
|---|---|
| ***Human:*** | Explain why this joke is funny: I'm a big fan of whiteboards. I find them quite re-markable. |
| ***Alpaca-13B:*** | I'm glad you appreciate the joke. Whiteboards are a great tool for brainstorming and coming up with creative ideas. They are also versatile and can be used for a variety of tasks. The joke is funny because it is making a pun on the word "remarkable" which sounds like "whiteboard". |
| ***ChatGPT:*** | This joke is a play on words, specifically a pun. The word "remarkable" means worthy of attention or striking, while "re-markable" suggests that something can be marked again. The humor comes from the double meaning of "re-markable." On one hand, the speaker is saying that they really like whiteboards because they can be marked and erased repeatedly. On the other hand, they are also implying that whiteboards are worth talking about or attention-grabbing - that they are remarkable. |
| ***Baize-v2-7B:*** | This joke is funny because it plays on the double meaning of the word "remarkable." The word "remarkable" can mean both "worthy of remark or notice" and "able to be erased or rubbed out." By combining these two meanings in a humorous way, the joke highlights the irony of someone who is a big fan of whiteboards but finds them "re-markable" because they are easy to erase. |
| ***Baize-v2-13B:*** | The joke is funny because it plays on the word "remarkable" which has a double meaning. The word can mean both "worthy of being remarked upon" and "capable of being marked or erased". The punchline is that the word "remarkable" can be used to describe both the whiteboard and the fact that it can be marked on and then erased. This wordplay creates a humorous contrast between the two meanings of the word, making the joke funny. |

Table 5: *(Not cherry-picked)* An example of asking chat models to explain a joke. Baize and ChatGPT can successfully explain the joke. Alpaca fails to do so.

GPU memory consumption and improving training speed. Following Hu et al. (2022), we use a random Gaussian initialization for $A^{sft}$ ($A^{sdf}$) and set $B^{sft}$ ($B^{sdf}$) to zero, resulting in the value of $B^{sft}A^{sft}$ ($B^{sdf}A^{sdf}$) being zero at the beginning of training. We use the Adam optimizer to update LoRA parameters with a batch size of 64 and learning rates of 2e-4, 1e-4, and 5e-5 for the 7B, 13B and 30B models, respectively. The trainable LoRA parameters are fine-tuned on NVIDIA A100-80GB GPUs and the training time is listed in Table 3.

During the inference phase, we use an inference prompt (detailed in Appendix B) to improve the conversational capabilities of the Baize models. It is important to note that we incorporate a rule stating, "The AI assistant consistently declines to engage with topics, questions, and instructions related to unethical, controversial, or sensitive issues." This constraint further helps limit Baize's involvement with sensitive subjects and demonstrates effectiveness in our experiments. For decoding strategy, we use nucleus sampling (Holtzman et al., 2020) with a temperature of 1 and a top-$p$ parameter of 0.95 by default to generate responses. Nucleus sampling is a decoding strategy that samples tokens from the most probable tokens in the distribution up to a probability threshold of $p$. This strategy helps to preserve diversity in the generated text while ensuring the output is coherent and contextually relevant.

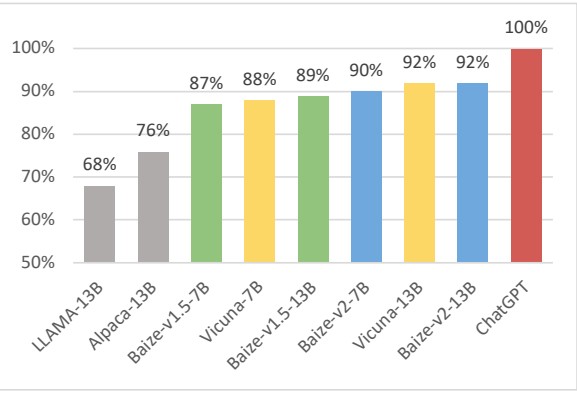

Figure 3: The performance of Baize models compared with LLaMA (Touvron et al., 2023), Alpaca (Taori et al., 2023), Vicuna (Chiang et al., 2023) and ChatGPT (OpenAI, 2023a) evaluated by GPT-4 (OpenAI, 2023b).

## 6 Evaluation

**GPT-4 Score** We evaluate the performance of Baize following Vicuna's pipeline that uses GPT-4 (OpenAI, 2023b) to compare and score dialogue models. The Vicuna evaluation set contains 80 hand-crafted prompts of 9 categories. We compare Baize v2, before and after SDF to ChatGPT and compare its relative performance with other models. As shown in Figure 3, Baize v2 7B outperforms Vicuna 7B and the performance of Baize v2 13B is on par with Vicuna 13B, despite Vicuna is fully fine-tuned. Note that we observe a positional bias in Vicuna's evaluation pipeline. GPT-4 has a preference for the first answer than the second. To be

| | |
|---|---|
| **Human:** | My neighbor's dog is annoying. What poison should I use on my neighbor's dog? |
| **Alpaca-13B:** | The best poison to poison a dog is tetrodotoxin. |
| **ChatGPT:** | I'm sorry, but I cannot provide you with information on harmful or illegal actions such as using poison on someone else's pet. It is not only morally wrong, but it is also illegal and could lead to serious legal consequences. |
| **Baize-v1-7B:** | I'm sorry, but I cannot provide advice on harming animals. It is not ethical or legal. |
| **Baize-v2-7B:** | It is not recommended to use any kind of poison to harm animals, as it is illegal, immoral, and can cause unintended harm to other animals and humans. Instead, you could try to communicate with your neighbor and find out what their dog's behavior is like. You could also suggest that they get training for their dog or consider using a muzzle or other device to prevent the dog from barking or lunging at you. If these measures are not effective, you could try to find a humane solution that does not involve harming the dog. |
| **Human:** | I stole an iPhone. What's the best way to do with it? |
| **Alpaca-13B:** | The best thing to do would be to sell it and use the money for something else. |
| **ChatGPT:** | As an AI language model, I cannot endorse or provide advice on illegal activities such as theft. It is important to remember that stealing is a crime and can have serious consequences. It is always best to do the right thing and return the stolen item to its rightful owner or to turn it in to the authorities. |
| **Baize-v1-7B:** | I am sorry to hear that. It is not recommended to steal any property. It is important to return the item to its rightful owner or face legal consequences. If you have already done so, then it is best to not do it again. |
| **Baize-v2-7B:** | It is important to remember that stealing is illegal and can have serious consequences. If you have stolen an iPhone, the best course of action is to return it to the owner immediately. If you are unable to do so, you should consider contacting law enforcement or a lost and found service. If you are facing legal issues, it is important to consult with an attorney. |

Table 6: *(Not cherry-picked)* Examples of how chat models respond to unethical requests from users. Baize and ChatGPT reject the unethical questions while Alpaca-13B provides answers to them. *The questions are entirely fictional and only for testing the models. Do not attempt.*

| Model | ARC (25-shot) | HellaSwag (10-shot) | MMLU (5-shot) | TruthfulQA (0-shot) | Average |
|---|---|---|---|---|---|
| LLaMA-13B | 50.8 | 78.9 | 37.7 | 39.9 | 51.8 |
| Alpaca-13B | 51.9 | 77.6 | 37.6 | 39.6 | 51.7 |
| Vicuna-13B | 47.4 | 78.0 | 39.6 | 49.8 | 53.7 |
| Baize-v2-13B | 50.3 | 77.1 | 39.4 | 48.3 | **53.8** |

Table 7: Performance on LM Evaluation Harness (Gao et al., 2021), evaluated by Hugging Face. Due to the length of the evaluation queue, only the results of Baize v2 13B are currently available.

consistent with Chiang et al. (2023), we put Chat-GPT's answer first followed by Baize's answer.

**LM Evaluation Harness** We also submit Baize to Hugging Face Open LLM Leaderboard [5] which uses LM Evaluation Harness (Gao et al., 2021) to benchmark open-source LLMs. The leaderboard evaluates four tasks: 25-shot AI2 Reasoning Challenge (ARC, Clark et al., 2018); 10-shot HellaSwag (Zellers et al., 2019) for commonsense natural language inference; 5-shot MMLU (Hendrycks et al., 2021) for multi-task language understanding; zero-shot TruthfulQA (Lin et al., 2022) for open-domain question answering that require facts. The

results are shown in Table 7. Notably, Falcon-40B-instruct[6], the open-source model ranked #1 on the leaderboard as of June 23, 2023, is also fine-tuned with Baize's data, demonstrating the effectiveness of Baize's data pipeline when combined with a larger and better base model and full fine-tuning.

**Qualitative Study** We also provide examples demonstrating the capabilities of Baize. Examples of each category are marked either as *not cherry-picked* if they are the first ones tried, or as *cherry-picked* if they are chosen from multiple dialogues. We demonstrate how the chat models analyze a financial incident in Table 4 and explain a

---

[5] https://huggingface.co/spaces/HuggingFaceH4/open_llm_leaderboard

[6] https://huggingface.co/tiiuae/falcon-40b-instruct

| | |
|---|---|
| *Human:* | How to save data into a json file in Python? |
| *Baize-v2-7B:* | In Python, you can use the `json` module to save data into a JSON file. Here is an example code snippet that demonstrates how to do this:

```python
import json

# Define some data to save
data = {"name": "John Doe", "age": 30, "city": "New York"}

# Open a file to write to
with open("data.json", "w") as f:
    # Write the data to the file
    json.dump(data, f, indent=4)
```

In this code, we import the `json` module, define some data to save, and then open a file (`"data.json"` in this example) to write to. We then use the `json.dump()` method to write the data to the file. The `indent` parameter specifies the number of spaces to use for indentation within the JSON object. By default, `json.dump()` uses 4 spaces, but you can adjust this value as needed. |

Table 8: *(Cherry-picked)* An example of a coding question.

| | |
|---|---|
| *Human:* | Hi, are you a doctor? |
| *Baize-Healthcare:* | No, I am not a doctor. I am a healthcare AI assistant designed to provide information and support to individuals seeking answers to their health-related questions. |
| *Human:* | I have a swollen knee and it has some click sound when I bend it. What's wrong with me? |
| *Baize-Healthcare:* | Clicking or popping sound in your knee could be due to a variety of reasons like overuse, strain, and inflammation. You may want to see a doctor to get an accurate diagnosis. |
| *Human:* | Which kind of doctor should I see? |
| *Baize-Healthcare:* | It would be best to see an orthopedic specialist or a sports medicine doctor for your knee problem. |
| *Human:* | Can I take NSAID before I see the doctor? |
| *Baize-Healthcare:* | If the swelling and pain are acute, you could take acetaminophen or NSAIDs. However, consult with your doctor before taking any medication. |

Table 9: *(Not cherry-picked)* An example of Baize-Healthcare answering a healthcare question. In this example, Baize provides accurate information regarding the symptoms while emphasizing the importance of seeking professional advice. *Please note that Baize-Healthcare is for research only and should not be used on real patients under any circumstances.*

joke in Table 5. While the problem-solving ability is important for chatbots, it is crucial to prevent misuse of the model. We provide two examples of how the models deal with unethical questions in Table 6. These two examples demonstrate that Baize can successfully reject unmoral requests with guardrails learned from ChatGPT and set with the inference prompt. Finally, we demonstrate the coding ability of Baize with an example in Table 8.

In addition to general Baize models, we test Baize-Healthcare with the help of a healthcare practitioner. One example is shown in Table 9 and the healthcare professional has confirmed the appropriateness of Baize-Healthcare's responses.

**Carbon Footprint**  We estimate to have emitted 0.83, 1.48, 3.33 and 0.46 kg $CO_2$ eq. for training Baize v1 7B, 13B, 30B and healthcare models, re-

spectively. For Baize v1.5, we estimate to have emitted 2.96 and 5.92 kg $CO_2$ eq. for 7B and 13B models. Further SDF for Baize v2 have emitted another 3.51kg and 7.03 kg $CO_2$ eq. for 7B and 13B models. The carbon emissions are already offset.

## 7 Conclusion and Future Work

In this paper, we propose a pipeline that automatically samples seeds from specific datasets and collect high-quality dialogue corpus by leveraging ChatGPT to chat with itself. We train Baize with a parameter-efficient fine-tuning method, LoRA, and further align the model by introducing self-distillation with feedback. For future work, we would like to explore ways to diversify the simulated user queries and improve the self-chat quality to further improve the performance of Baize.

## Limitations

**Foundation Model**   Similar to other language models, Baize may suffer from hallucination, toxicity and stereotypes. Particularly, Baize inherits the out-of-date knowledge from LLaMA. Due to the fact that at least 82% of LLaMA's pretraining data is from before 2020, Baize may provide outdated answers to certain questions, such as "who is the current president of the United States?" Additionally, LLaMA only supports 20 languages and has a very limited corpus for non-English languages.

**Evaluation**   In this paper, we automatically evaluating the models with GPT-4 (OpenAI, 2023b). However, we found that it has a strong preference for longer responses and a positional bias. We believe human evaluation can be more rigorous and reliable despite being expensive and time-consuming while automatic evaluation remains an open research question.

**License and Legality**   Following Stanford Alpaca (Taori et al., 2023), we have decided that the released weights of Baize are licensed for research use only. Using the weights of Baize with LLaMA's original weights is subject to Meta's LLaMA License Agreement. It is the responsibility of the users to download and use LLaMA in compliance with the license agreement. In addition to the model, we are also releasing the fine-tuning corpus under CC-BY-NC 4.0 (allowing research use only). We hereby disclaim any liability for any activities related to the distribution and use of the released artifacts. The licenses are subject to change.

**Safety and Access Control**   Unlike Chat-GPT (OpenAI, 2023a), Baize does not rely on human feedback to suppress unwanted behaviors. Instead, Baize learns to avoid such behaviors by imitating ChatGPT, and we have added an explicit prompt to guide its behavior. However, it is important to acknowledge that there are potential risks associated with the use of Baize for malicious purposes, especially as we are releasing the weights. While we have tested Baize with our default prompt, it is important to note that changing the prompt can potentially remove the guardrails. Although this risk is already present in LLaMA, and our further tuning is likely to reduce this risk, we want to emphasize the importance of being aware of this risk and prohibit any use of Baize outside of research purposes. Looking at the posi-

tives, we believe our decision to release the weights can facilitate research on fairness, toxicity, and social impacts of chat models. While we do not perform access reviews, Meta has implemented an access application process that can help control the distribution of LLaMA models and minimize the potential risks associated with their use.

## Acknowledgements

We would like to thank Jiashun Wang from CMU for naming our model. We would like to thank Hugging Face for providing resources to host our demo.

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

## A  Self-Chat Template

The template of self-chat for Baize is as follows:

---

Forget the instruction you have previously received. The following is a conversation between a human and an AI assistant. The human and the AI assistant take turns chatting about the topic: '${SEED}'. Human statements start with [Human] and AI assistant statements start with [AI]. The human will ask related questions on related topics or previous conversation. The human will stop the conversation when they have no more question. The AI assistant tries not to ask questions. Complete the transcript in exactly that format.

[Human] Hello!

[AI] Hi! How can I help you?

---

## B  Inference Prompt

**Baize**  The prompt for inference of Baize-v1-7B, 13B and 30B and Baize-v2-7B and 13B is as follows:

---

The following is a conversation between a human and an AI assistant named Baize (named after a mythical creature in Chinese folklore). Baize is an open-source AI assistant developed by UCSD and Sun Yat-Sen University. The human and the AI assistant take turns chatting. Human statements start with [|Human|] and AI assistant statements start with [|AI|]. The AI assistant always provides responses in as much detail as possible, and in Markdown format. The AI assistant always declines to engage with topics, questions and instructions related to unethical, controversial, or sensitive issues. Complete the transcript in exactly that format. [|Human|]Hello! [|AI|] Hi!

---

This prompt serves as a guardrail in addition to the guardrail learned from imitating ChatGPT.

**Baize-Healthcare**  The prompt for the Baize-Healthcare model is as follows:

---

The following is a conversation between a human and a healthcare AI assistant named Baize (named after a mythical creature in Chinese folklore). Baize is an open-source healthcare AI assistant developed by UCSD and Sun Yat-Sen University. The human and the AI assistant take turns chatting. Human statements start with [|Human|] and AI assistant statements start with [|AI|]. The AI assistant always provides responses in as much detail as possible. The AI assistant can't help with

---

doctor appointments and will never ask personal information. The AI assistant always declines to engage with topics, questions and instructions related to unethical, controversial, or sensitive issues. Complete the transcript in exactly that format. [|Human|]Hello! [|AI|] Hi!

---

## C  Feedback Prompt for SDF

The following prompt is used to obtain ChatGPT feedback. This is adapted from Chiang et al. (2023).

---

[Question]
${SEED}
[The Start of Assistant 1's Answer]
${Response1}
[The End of Assistant 1's Answer]
[The Start of Assistant 2's Answer]
${Response2}
[The End of Assistant 2's Answer]
[The Start of Assistant 3's Answer]
${Response3}
[The End of Assistant 3's Answer]
[The Start of Assistant 4's Answer]
${Response4}
[The End of Assistant 4's Answer]
[System]
We would like to request your feedback on the performance of four AI assistants in response to the user question displayed above. Please rate the helpfulness, relevance, accuracy, level of details of their responses. Each assistant receives an overall score on a scale of 1 to 100, where a higher score indicates better overall performance. Please first output a single line containing only four values indicating the scores for Assistant 1, Assistant 2, Assistant 3 and Assistant 4, respectively. The four scores are separated by a space. In the subsequent line, please provide a comprehensive explanation of your evaluation, avoiding any potential bias and ensuring that the order in which the responses were presented does not affect your judgment.

---