# OpenReview forum: "Baize: An Open-Source Chat Model with Parameter-Efficient Tuning on Self-Chat Data"
_EMNLP/2023/Conference — EMNLP 2023 Main_

### Official Review · Reviewer_Guni · 2023-08-05

**Soundness:** 2

**Excitement:**

2: Mediocre: This paper makes marginal contributions (vs non-contemporaneous work), so I would rather not see it in the conference.

**Paper Topic And Main Contributions:**

- This paper proposes a pipeline that automatically generates a multi-turn chat corpus using ChatGPT.
- Based on collected dataset, the authors present a chatting model named Baize, trained with parameter-efficient tuning based on LLaMA.
- They additionally apply reinforcement learning with self-feedback to improve the performance of Baize.

**Reasons To Accept:**

This paper presents valuable resources (e.g. dataset and model) for research community.

**Reasons To Reject:**

- The novelty of the proposed pipeline is limited, which has been already explored in several works [1], [2]. It is popular approach to generate dataset using LLMs and train smaller LMs on the collected dataset nowadays, and there is no other contributions to be considered in the proposed framework.
- There is no quality control (e.g. filtering) or manual evaluation on the quality of data, which is collected through automatic model-based approach.
- It is unclear which chatting model the authors aim to develop, and the motivation for developing such model is also unclear.

[1] Symbolic Knowledge Distillation: from General Language Models to Commonsense Models (NAACL 2022)
[2] SODA: Million-scale Dialogue Distillation with Social Commonsense Contextualization (ACL 2023 Findings)

**Reproducibility:**

4: Could mostly reproduce the results, but there may be some variation because of sample variance or minor variations in their interpretation of the protocol or method.

**Reviewer Confidence:**

4: Quite sure. I tried to check the important points carefully. It's unlikely, though conceivable, that I missed something that should affect my ratings.

---

> ### Author Rebuttal · Authors · 2023-08-27
>
> Thank you for your comments.
>
> 1. **Re. Novelty**: We understand that the reviewers may be lukewarm, as in the past months, there are so many open-source instruction-following models made available. However, we would like to point out that:
> - Baize is one of the *first attempts* to distill proprietary models into LLaMA.
> - Baize does not rely on external user data and can generate high-quality domain-specific dialogues.
> - Our work focuses on evaluating the effectiveness of the self-chat pipeline, which can easily obtain high-quality dialog data for a specific domain. Our evaluation has clearly demonstrated that the Baize pipeline obtains improvement over Self-Instruct and the baseline, and the final performance is on par with Vicuna.
> - Our work verifies the feasibility of using only one GPU to fine-tune an LLM (with a limited number of instruction data and LoRA). This was previously unknown before Baize.
> - We would like to highlight our contribution of **Self-Distillation with Feedback (SDF)**, a simple yet effective way to further improve model performance after supervised fine-tuning (SFT). This contribution seems to be overlooked by the reviewers.
> - Baize has inspired many works and the pipeline has helped develop models that perform better, including Falcon, Fauno and UltraLM, not to mention many open-source models available on Hugging Face Model Hub.
> 2. **Re. Quality Control**: As ChatGPT/GPT-4 overall generates high-quality text, it is very hard to apply any rule-based filtering or even manual annotations (this is also why detecting ChatGPT-generated text is hard) to further improve the quality of the collected data. However, there are certain ways to further improve the quality of data automatically, e.g., best-of-n selection, answer ensemble (e.g., LLM-Blender [1]), chain-of-thought, etc. These techniques come with the price of more API calls. We believe this is a trade-off to make depending on the application scenario. In our work, we apply self-distillation with feedback (SDF) as another measure to further improve the performance by introducing real-time feedback from ChatGPT.
> 3. **Re. Motivation**: We demonstrate the feasibility of using self-chat and SDF to obtain a chat model for general domain (Baize v1, v2) and for a specific domain (Baize-Healthcare).
>
> ----
> [1] Jiang et al., LLM-Blender: Ensembling Large Language Models with Pairwise Ranking and Generative Fusion

---

### Official Review · Reviewer_iPyR · 2023-08-05

**Soundness:** 3

**Excitement:**

3: Ambivalent: It has merits (e.g., it reports state-of-the-art results, the idea is nice), but there are key weaknesses (e.g., it describes incremental work), and it can significantly benefit from another round of revision. However, I won't object to accepting it if my co-reviewers champion it.

**Paper Topic And Main Contributions:**

This paper proposes a framework for efficiently performing Instruction-tuning on large language models (LLMs), utilizing ChatGPT. The framework involves conducting Self-chat using ChatGPT for collecting dialogues for the first stage of training. Subsequently, the obtained model is further trained using ChatGPT's Feedback as an alternative to RLHF, successfully enhancing the model's performance.


**Reasons To Accept:**

- The paper proposes an efficient and practical method for training high-quality LLM without the intervention of actual humans. It is considered an efficient and reproducible method compared to Vicuna, which is learned from actual human interactions with ShareGPT.
- The evaluation has been conducted at each step of the proposed method, clearly demonstrating the utility of each part.
- The paper is written clearly throughout, making it easy for readers to understand.


**Reasons To Reject:**

- The evaluation is limited to automated assessments, including those based on GPT, and lacks human evaluations.
- As the learning is based on ChatGPT, the experimental conditions seem to favor the proposed method that uses the same model for training data; however, there is no discussion regarding this aspect.
- Although multiple generation examples are shown, a detailed analysis of the model's performance is lacking, and the differences compared to other models like Vicuna are not clearly illustrated.

**Reproducibility:**

3: Could reproduce the results with some difficulty. The settings of parameters are underspecified or subjectively determined; the training/evaluation data are not widely available.

**Reviewer Confidence:**

3: Pretty sure, but there's a chance I missed something. Although I have a good feel for this area in general, I did not carefully check the paper's details, e.g., the math, experimental design, or novelty.

---

> ### Author Rebuttal · Authors · 2023-08-27
>
> We would like to thank the reviewer for your insightful comments.
>
> 1. **Re. Evaluation**: Additional benchmark results are available on MT-Bench and Alpaca Eval (unfortunately EMNLP does not allow authors to add links in the rebuttal). Both benchmarks have done human evaluation correlation studies to verify the effectiveness of using GPT-4 to evaluate. **Please refer to our response to Reviewer ftdW about the performance of Baize and how to improve it.**
> 2. **Re. Favored comparison**: We use ChatGPT (gpt-3.5-turbo) for both supervised fine-tuning (SFT) and self-distillation with feedback (SDF). We use GPT-4 for evaluation. On the contrary, Vicuna is trained with a mix of ChatGPT and GPT-4 responses, thus the evaluation pipeline actually favors Vicuna, instead of Baize.

---

### Official Review · Reviewer_ftdW · 2023-08-10

**Typos Grammar Style And Presentation Improvements:** N/A
**Soundness:** 3

**Excitement:**

4: Strong: This paper deepens the understanding of some phenomenon or lowers the barriers to an existing research direction.

**Missing References:**

N/A

**Paper Topic And Main Contributions:**

The paper proposed an efficient pipeline (low cost, low compute) to imitate proprietary chat models. The paper utilized self-chat and self-distill techniques to generate chat data and human feedback data, in comparision with SFT and RLHF proposed by OpenAI's InstructGPT. Self-chat avoids human annotation, and self-distill avoids human feedback. Authors then trained a series of models based on such pipeline, such as (Baize v1, Baize v1.5 Baize v2) with (7B, 13B) parameters.  The self-distill process further added an extra LoRA module to the base model. The paper evaluate Baize using LM Evaluation Harness library and GPT-4 evaluator. The pipeline highlights efficency.

**Questions For The Authors:**

Have you ever tried using full parameter finetuning, what is the difference between full-parameter finetuning and LoRA on your data?

The generated data may contain hallucination, finetuning on such data seems to cause model generate random response. Have you ever tried some methods to avoid that?

Do you have any future plan to improve Baize's pipeline?

Do you think if you have a base model of ChatGPT (gpt-3.5), using self-chat and self-distill can yield a similar model with ChatGPT? If not, what is the reason?

Are you confident on your proposed pipeline?

**Reasons To Accept:**

The paper proposed an efficient pipeline to imitate state-of-the-art proprietary chat models such as ChatGPT.

The paper proposed a novel self-chat technique, which avoids human annotation.

The paper proposed a novel self-distill technique, which avoids human feedback.

The pipeline highlights efficency.

**Reasons To Reject:**

I noticed that author stated that "Different from these attempts, our work focuses on developing an affordable and reproducible pipeline to efficiently tune a general-purpose language model for multi-turn chat." Despite this, Baize's current results are not competitive on leaderboards like Huggingface OpenLLM. Considering running Baize requires the same resource as other high-ranking Llama-13B models, and the author failed to point out any directions to further make Baize competitive with other 13B models. So I suspect that the proposed pipeline not work. I think authors should further improve the proposed pipeline to get a competitive result.

The generated data may contain hallucination, finetuning on such data seems to cause model generate random response.

**Reproducibility:**

5: Could easily reproduce the results.

**Reviewer Confidence:**

4: Quite sure. I tried to check the important points carefully. It's unlikely, though conceivable, that I missed something that should affect my ratings.

---

> ### Author Rebuttal · Authors · 2023-08-27
>
> Thank you for your insightful comments.
>
> 1. **Re. Performance on leaderboards**:
> - Additional evaluation results on MT-Bench and Alpaca Eva are available. The performance of Baize v2 is on par with Vicuna v1.3, despite Vicuna uses a larger amount of ShareGPT data (which is a mix of ChatGPT and GPT-4) and is tuned with full parameters.
> - Vicuna is a concurrent work, while many community models were published after Baize. As we are glad to see the prosperity of the open-source community, it is unfair to evaluate the value of Baize by comparing it with newer models. Baize is nearly 5 months old and we cannot publish new models and update the results on the leaderboards due to EMNLP anonymity period restrictions.
> - Baize has inspired many works and the pipeline has helped develop models that perform better, including Falcon, Fauno and UltraLM [1], not to mention many open-source models available on Hugging Face Model Hub.
>
> To improve the performance of Baize is easy:
> - One can scale up the self-chat pipeline to cover more topics with more diverse seed datasets, e.g., UltraLM [1].
> - Apply fine-tuning with full parameters instead of parameter-efficient fine-tuning, e.g., UltraLM [1], Vicuna.
> - Use GPT-4 responses/feedback instead of ChatGPT’s, e.g., GPT-4-LLM [2]. We intentionally did not use GPT-4 responses for Baize as we are evaluating Baize with GPT-4. Using GPT-4 responses to train the model could lead to an unfair advantage when evaluated with GPT-4.
> - Replace the base model with LLaMA 2, e.g., Vicuna v1.5
>
> 2. **Re. Full fine-tuning**: We intended to verify the effectiveness of Baize under a low-resource setting. We show that with limited computation resources and a low budget of API calls, one can efficiently tune a chat model. To answer your question, Sun et al. [3] compares LoRA tuned models with full fine-tuning. Although the LoRA tuned models perform worse than full fine-tuning, there are some merits, including reduced VRAM usage, faster tuning speed, and easy weight distribution. Although we did not have enough resources to do full fine-tuning, the community has done full fine-tuning with Baize data. For example, as we mentioned in Ln. 362, Falcon-40B-instruct is trained with full fine-tuning on Baize v1 data.
> 3. **Re. Hallucinations**: Indeed, the generated data could introduce hallucinations. However, a later study (Zhou et al.) [4] concludes that the underlying base model (e.g., LLaMA) may be more important than the instructions used to fine-tune the model (e.g., Baize data). To mitigate that, we introduce Self-Distillation with Feedback (SDF). As shown in Figure 3, online SDF can improve the performance of Baize and we assume it can help reduce hallucinations, as it asks ChatGPT to check the answers of Baize and provide supervision for improvement.
> 4. **Re. Future plans**: For future work, we plan to develop a more efficient and effective alignment technique than SDF and RLHF. Meanwhile, we believe the performance of Baize can be easily improved, as we highlighted before.
> 5. **Re. Instruction-GPT as base model**: Vicuna v1.5 has shown that with a better base model (i.e., LLaMA 2), the performance can be very close to ChatGPT. As InstructGPT (the base model of ChatGPT) still holds an advantage against LLaMA 2, we predict the performance of such a hypothetical model can be close to ChatGPT. However, it is unlikely the model will be identical to ChatGPT, as the number and domain diversity of dialogs could be a bottleneck for transferring all knowledge to this hypothetical model. We believe this will match the observation on knowledge distillation, where the student model can match the teacher performance on in-domain data but will not generalize as well as the teacher model on out-of-domain data.
>
> ----
>
> - [1] Ding et al., Enhancing Chat Language Models by Scaling High-quality Instructional Conversations
> - [2] Peng et al., Instruction Tuning with GPT-4
> - [3] Sun et al., A Comparative Study between Full-Parameter and LoRA-based Fine-Tuning on Chinese Instruction Data for Instruction Following Large Language Model
> - [4] Zhou et al., LIMA: Less Is More for Alignment

---

### Meta-Review · Area_Chair_pxcp · 2023-10-06

**Recommendation:** 3

**Metareview:**

The novelty of the proposed pipeline is questioned, with references to prior works exploring similar approaches. The paper needs to clarify its unique contributions and distinguish itself from existing research. In addition, there is a lack of human evaluation. However, reviewers acknowlege the contributions made by the approach outlined in the paper. Given the reviewers, the author rebuttal and following discusssion, I recommend the paper be accepted to Findings track.

---

### Decision · Program_Chairs · 2023-10-07

**Decision:**

Accept-Main

**Comment:**

The novelty of the proposed pipeline is questioned, with references to prior works exploring similar approaches. The paper needs to clarify its unique contributions and distinguish itself from existing research. In addition, there is a lack of human evaluation. However, reviewers acknowlege the contributions made by the approach outlined in the paper. Given the reviewers, the author rebuttal and following discusssion, I recommend the paper be accepted to Findings track.